# Propagation of ground penetrating radar waves in Chinese coals

**Duo Zhang**[1,2,3]*, **Rui Tang**[1,2], **Hu Wen**[1,2,3], **Shixing Fan**[1,2]

**1** School of Safety Science and Engineering, Xi'an University of Science and Technology, Xi'an Shaanxi, China, **2** Key Laboratory of Mine and Disaster Prevention and Control of Ministry of Education, Xi'an University of Science and Technology, Xi'an Shaanxi, China, **3** Postdoctoral Mobile station of Mining Engineering, Xi'an University of Science and Technology, Xi'an Shaanxi, China

* zhangd@xust.edu.cn

## Abstract

This paper reports an experimental study on the electrical properties of five coal samples taken from various Chinese coal mines. The dielectric permittivity and specific resistivity of grinded coal samples subjected to electromagnetic (EM) fields in a wide frequency spectrum were determined. Based on the experimental data, a set of approximating equations of the change in electric properties the 100–1000 MHz frequency region was obtained. These equations, along with EM equations for EM speed and attenuation, were used for modeling and studying radar-wave propagation in a coal seam and radar-wave reflection from the body of miners trapped in collapsed tunnels. The modeling concept assumes that a radar transducer with the dominating frequency of 500 MHz is lowered through a vertical or inclined rescue borehole to the depth of the coal seam. It is assumed that only the miner is present in the part of the tunnel that did not collapse. Thus, in the path of the radar wave from the transducer to the human body, only one geological interface reflecting the radar signals is present (coal-air) and one is connected with human body. The human (acting as the reflector) can be located at various distances from the tunnel face; this factor was included in the analyzed geometrical model. Based on the modeling results of different thickness coal seams (2, 3, 4, 5, 6, 7, 8m), conclude that a radar wave reflected from a human body can be reliably measured, when the distance between the human and the transducer is not exceed 8m.

**Data Availability Statement:** All relevant data are within the paper.

**Funding:** Funding: The project was supported by the National Natural Science Foundation of China (grant numbers 51904234 and 51974240) and the Key R & D Program of Shaanxi Provincial (grant number 2017ZDCXL-GY-01-02-03).

## 1. Introduction

In China, coal appears to be the main energy source for the near future [1, 2]. Additionally, it constitutes a large proportion of the global primary energy sources [3–5]. Accordingly, safety issues in coal mining processes have always been a research hotspot: coal mine emergency rescue remains as the last barrier to safety and security.

In recent years, with advancements in science, technology, management, and quality of workers, the safety of coal mines has clearly improved. The total number of accidents and death tolls have dropped significantly. However, major accidents in coal mines continue to

**Competing interests:** The authors have declared that no competing interests exist.

occur occasionally. For example, two particularly significant coal mine accidents and eight major accidents occurred in 2016 [6]. Coal mine accidents inflict large numbers of casualties, along with severe economic losses and adverse social impacts [7]. The earlier the location and condition of the people caught in mining accidents are known, the greater the possibility of them being rescued. The ground rescue method must be considered on encountering the following two scenarios: (a) the roadway could not be used because of coal or rock fall, high temperature, and presence of water, among others adverse conditions. (b) the lives of the rescue workers are threatened. To date, some successful drilling rescue cases have been recorded. For example, a water permeable accident occurred in the Kuixi Coal Mine in USA in July 2002. A mine collapse accident occurred in San Jose copper mine in Chile in August 2010 [8]. A mine collapse accident occurred in Yurong gypsum mine in China in December 2015 [9]. Long-distance drilling could be deflected during construction under the current technical conditions (Fig 1). Under these circumstances, life information detection technologies, such as video or audio, infrared, and gas, would not be available because they cannot penetrate the coal seam [10]. Among numerous non-contact life information detection technologies, ground penetrating radar (GPR) radar is considered to as the most effective because of its high resolution, deep penetration, and low power radiation, making it suitable for life rescue systems [11, 12]; thus it has been used for life detection and rescue [13]. Many works have been conducted on life detection based on radar [14–18]. Based on eigenvalues using UWB impulse radar remote sensing, Minhhuy studied the heart rate extraction [19]. Using UWB impulse radar, Liang et al studied the improved denoising method for through-wall vital sign detection [20]. Therefore, we believe that the GPR radar technology could be the direction of future development of life information detection through coal seams. This study shows that the propagation of GPR radar waves in coal, and the related influencing factors have very important scientific and practical significance.

Numerous scholars have extensively studied the electromagnetic properties of coal and the propagation of electromagnetic waves in coal. Fan et al. studied the dielectric properties of coals in the low THz frequency region [21]. Fan et al. studied the dielectric properties of coal in the THz frequency region of 100–500 GHz [22]. Qin and Wei studied the relative dielectric constant sweep measurement based on a vector network analyzer [23]. Brach studied the real part of the permittivity of a series of coals as a function of frequency (100 Hz–$1.3 \times 10^7$ Hz) and temperature (200–400 K); the results showed that certain parameters are related to the microstructure of the carbonaceous phase [24]. Marland et al. tested the dielectric properties of coal

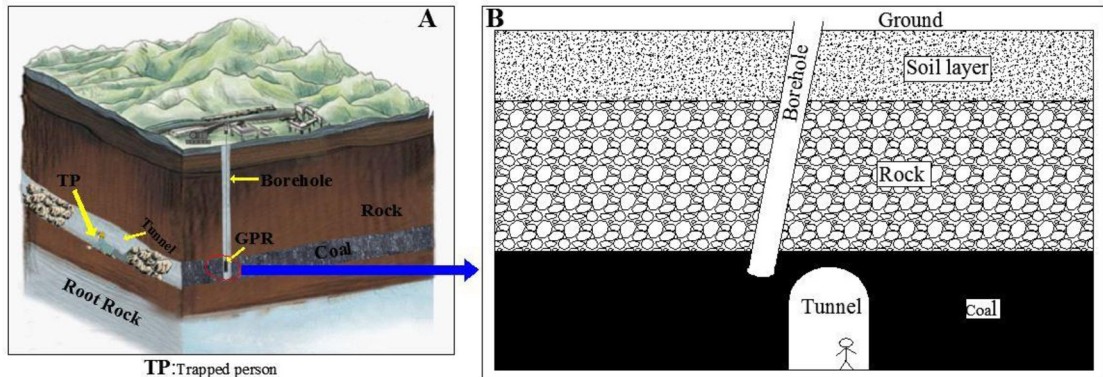

**Fig 1. Schematic concept of and application of borehole GPR method for location of miner trapped in underground tunnel in coal mine. A** is a three-dimensional diagram of rescue drilling during the tunnel collapse; **B** is a schematic diagram of the drilling offset.

using the microwave cavity method in the laboratory, and qualitatively discussed the effects of temperature, frequency, coal rank, minerals, and water content on the coal dielectric properties [25]. Valentina studied the influence of polyethylene terephthalate on the carbonization of bituminous coals and on the modification of their electric and dielectric properties [26]. The relationship between interior moisture distribution and coal dielectric properties has been constructed by Wang. Furthermore, the characteristics of permittivity variation with moisture content has been studied [27]. Peng et al studied the dielectric characteristics of an Indonesian low-rank coal by determining its permittivity from 21°C to approximately 1000 °C at 915 MHz and 2450 MHz in argon [28]. Tao et al. analyzed the effect of sulfur content on the dielectric properties of coal [29]. Xu et al. studied the influence of the crystallite structure on the dielectric properties of coal chars [30]. Liu studied the effect of coal rank on the dielectric properties through structural differences in various coal chars [31]. Emslie and Lagace. put forward the model of a coal seam waveguide, for the first time, when studying radio wave communication in coal mines [32]. Hill derived the theoretical formula of electromagnetic wave propagation under the layered model and studied the propagation of electromagnetic waves in uneven coal seams [33, 34]. Greenfield used finite difference simulations to analyze the attenuation characteristics of electromagnetic waves in discontinuous coal seams [35]. Luo et al. analyzed the effect of electrical properties of coal seams on the attenuation characteristics of electromagnetic waves [36–39]. Zhu et al. reviewed the research progress on the measurement technique of coal dielectric constants and analyzed the dielectric models of several coals [40]. Clearly, only a few have investigated the electrical parameters of coals of different ranks in the frequency range of 100–1000 MHz.

In this paper, therefore, we report an experimental study on the dielectric properties of five coal samples of different ranks, with frequencies ranging from 100–1000 MHz. Furthermore, we construct the approximation equations for propagation of radar waves in coal, and simulate and solve the propagation process. Finally, the effective range of penetration of radar waves in coal is given.

## 2. Experiment and methods

### 2.1. Measurement of dielectric properties

In this study, five different coal samples were used based on their ranks: lignite (1#), Dananhu coal mine, Xinjiang; long flame coal (2#), Tangjiahui coal mine, Inner Mongolia; gas coal (3#), Baodian coal mine, Shandong; lean coal (4#), Sangshuping coal mine, Shaanxi; anthracite (5#), Huangyanhui coal mine, Shanxi. The proximate analysis and elemental analysis results are provided in Table 1. Fresh coal samples, which were not subjected to water injection, spraying, and other treatments were sealed and packed with multilayer plastic and nylon bags, and transported to the laboratory. Each experiment was conducted four times to ensure reproducibility. The samples were prepared as follows. A fresh coal sample was crushed, and pulverized coal having a diameter of less than 0.074 mm was screened out. A certain amount of pulverized

**Table 1. Proximate and elemental analyses of coal.**

| Coal | Proximate analysis (%) | | | | Elemental analysis (%) | | | | | Porosity (%) |
|---|---|---|---|---|---|---|---|---|---|---|
| | $M_{ad}$ | $A_{ad}$ | $V_{ad}$ | $FC_{ad}$ | C | H | O | N | S | |
| 1# | 8.80 | 36.62 | 24.68 | 29.90 | 38.94 | 4.01 | 22.00 | 1.15 | 0.27 | 10.11 |
| 2# | 5.77 | 11.93 | 32.32 | 49.98 | 64.86 | 5.63 | 19.20 | 1.73 | 0.58 | 13.34 |
| 3# | 2.33 | 15.92 | 35.45 | 46.30 | 63.92 | 4.85 | 17.40 | 2.37 | 0.51 | 6.16 |
| 4# | 0.53 | 24.49 | 12.95 | 62.03 | 62.51 | 3.76 | 17.00 | 1.61 | 3.41 | 9.19 |
| 5# | 1.20 | 8.84 | 7.85 | 82.11 | 81.82 | 3.79 | 4.90 | 0.85 | 1.49 | 16.20 |

coal was weighed into a tableting machine to prepare a circular coal sample piece of diameter 1.28 cm and thickness 1.0 mm. Then conductive silver colloid was evenly coated on both sides of the coal sample, and could be tested after the silver gel dries. The preparation process of the sample is illustrated in Fig 2.

The Concept-80 broadband dielectric spectrum test system was employed in the equipment, which mainly included the control software, an E4991A impedance analyzer and test fixture (Fig 3).

## 2.2. Propagation equations of radar waves in coal

**2.2.1. Dielectric properties of tested coal samples.** The relative dielectric constant values of the five tested coals are shown in Fig 4, and the resistivity values are shown in Fig 5.

Fig 4 shows the trend of the relative dielectric constants of the five coal samples were similar. They decreased first and then increased with the increasing test frequency. The minimum values of the dielectric constant occurred at approximately 300 MHz, whereas the largest values occurred at approximately 1000 MHz. At the same test frequency, lean coal had the highest relative dielectric constant, followed by lignite, gas coal, long flame coal, and anthracite (Fig 4 (F)). The lignite contained a large number of hydrophilic polar functional groups, which could adsorb a large amount of intrinsic moisture. Polar functional groups and moisture had a high relative dielectric constant. Therefore, the dielectric constant of lignite was higher. The order of relative dielectric constants of the other four ranks of coal should be determined based on the carbon content in coal. The relative dielectric constant of lean coal was higher than that of lignite because it was influenced by other factors, such as porosity [24]. The relative dielectric constants and frequencies were fitted and were found to satisfy the cubic polynomial relationship. The results are given by Eq (1), and the correlation coefficients are listed in Table 2.

$$\varepsilon = C_3 f^3 + C_2 f^2 + C_1 f + C_0, \left( f \in [100MHz, 1000MHz] \right) \tag{1}$$

where $\varepsilon$ is the relative dielectric permittivity, $f$ denotes the frequency, and $C_i$ is a coefficient ($i = 0, 1, 2, 3$).

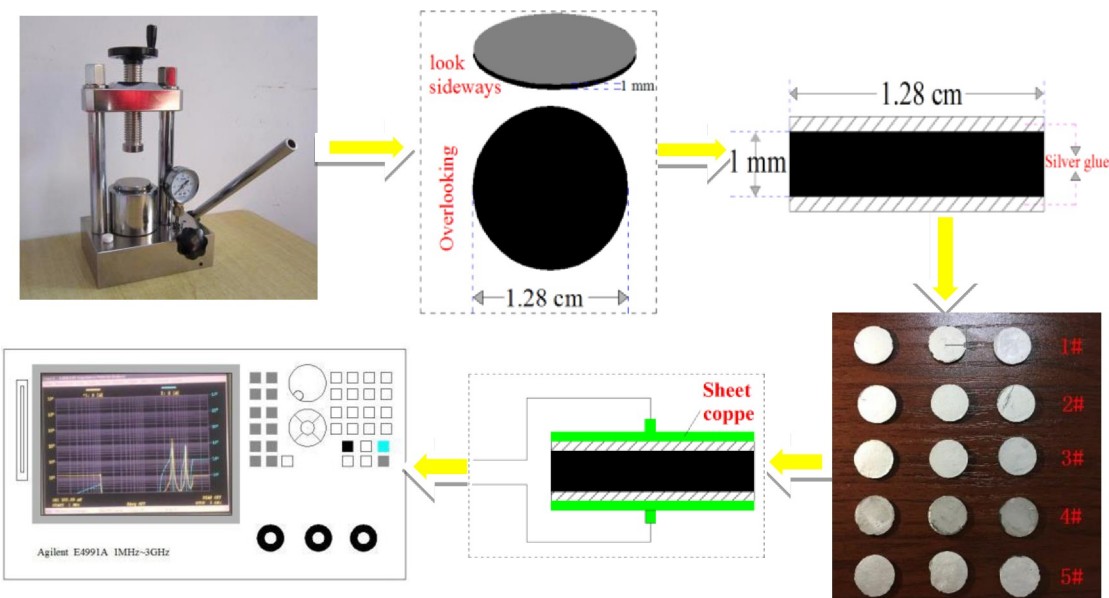

**Fig 2. Flow chart of processing test of coal samples.**

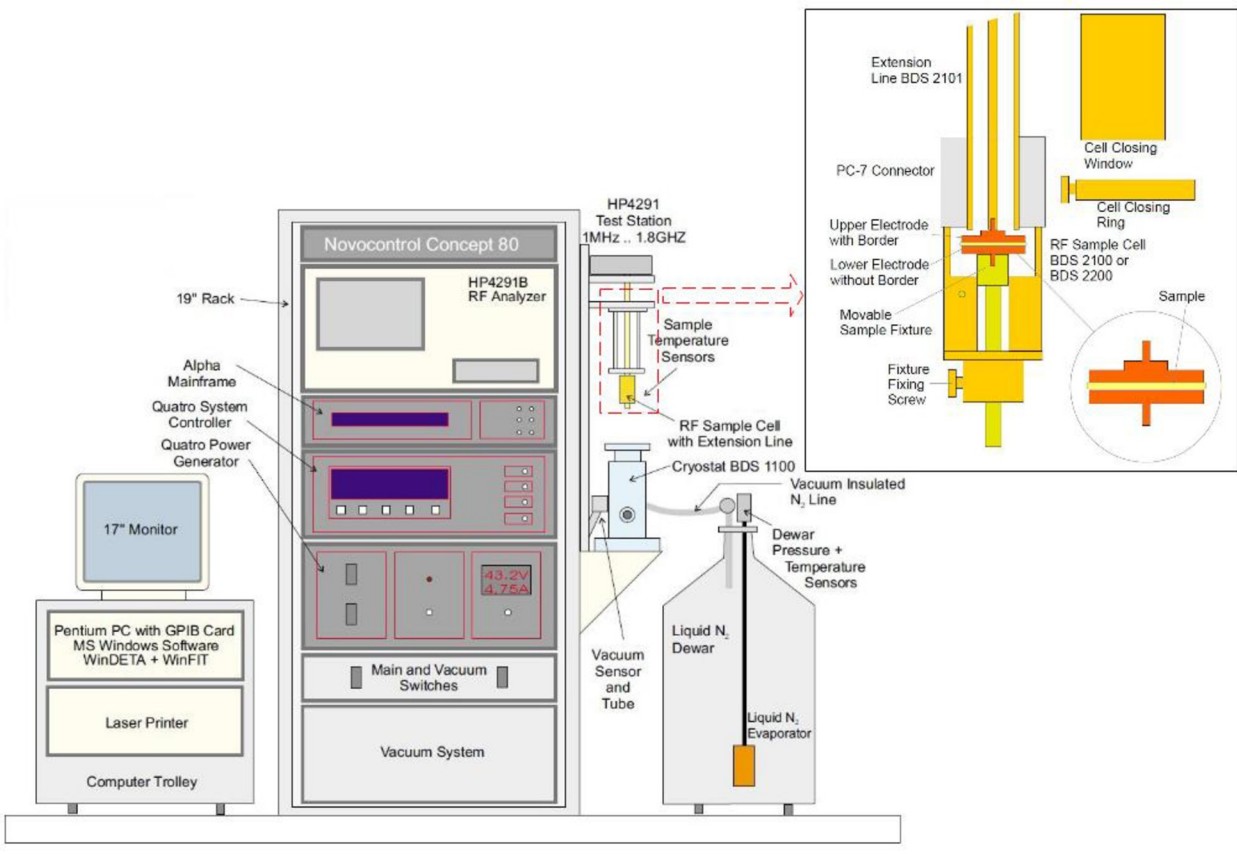

**Fig 3. Concept-80 broadband dielectric spectrum test system.**

Fig 7 shows the resistivity of coal with a different degree of metamorphism had an overall decreasing trend with the increasing test frequency. A convex transformation point occurred near 500 MHz, which could divide the trend of resistivity with frequency into two stages. It was found, through fitting, that part I conformed to a cubic polynomial and part II conformed to a quadratic polynomial. The results are given in Eq (2) and Table 3. Beyond 700 MHz, the resistivity of the tested coals tended to be flat and the differences between them were small. As the test frequency increases, the electrons in a bundled state of the coal structure group become freely excited electrons, causing a significant increase in free radicals (i.e., unpaired electrons) of coals; therefore, the resistivity would decrease. Previous studies have shown that anthracite is a resistive absorbing material [41], and the resistivity of its mineral impurities is higher than that of organic matter [42]. These factors could make the resistivity of anthracite relatively large.

$$\rho=\begin{cases} C_{13}f^3 + C_{12}f^2 + C_{11}f + C_{10}, \ (f \in [100MHz, 500MHz]) \\ C_{22}f^2 + C_{21}f + C_{20}, (f \in (500MHz, 1000MHz]) \end{cases} \tag{2}$$

**2.2.2. Equation of radar wave propagation in coal.** To simplify the model, we assumed that the coal body was a homogeneous medium. Because radar waves are a type of electromagnetic waves, their propagation velocity in coal could be expressed by Eq (3) [43] and Fig 6, and

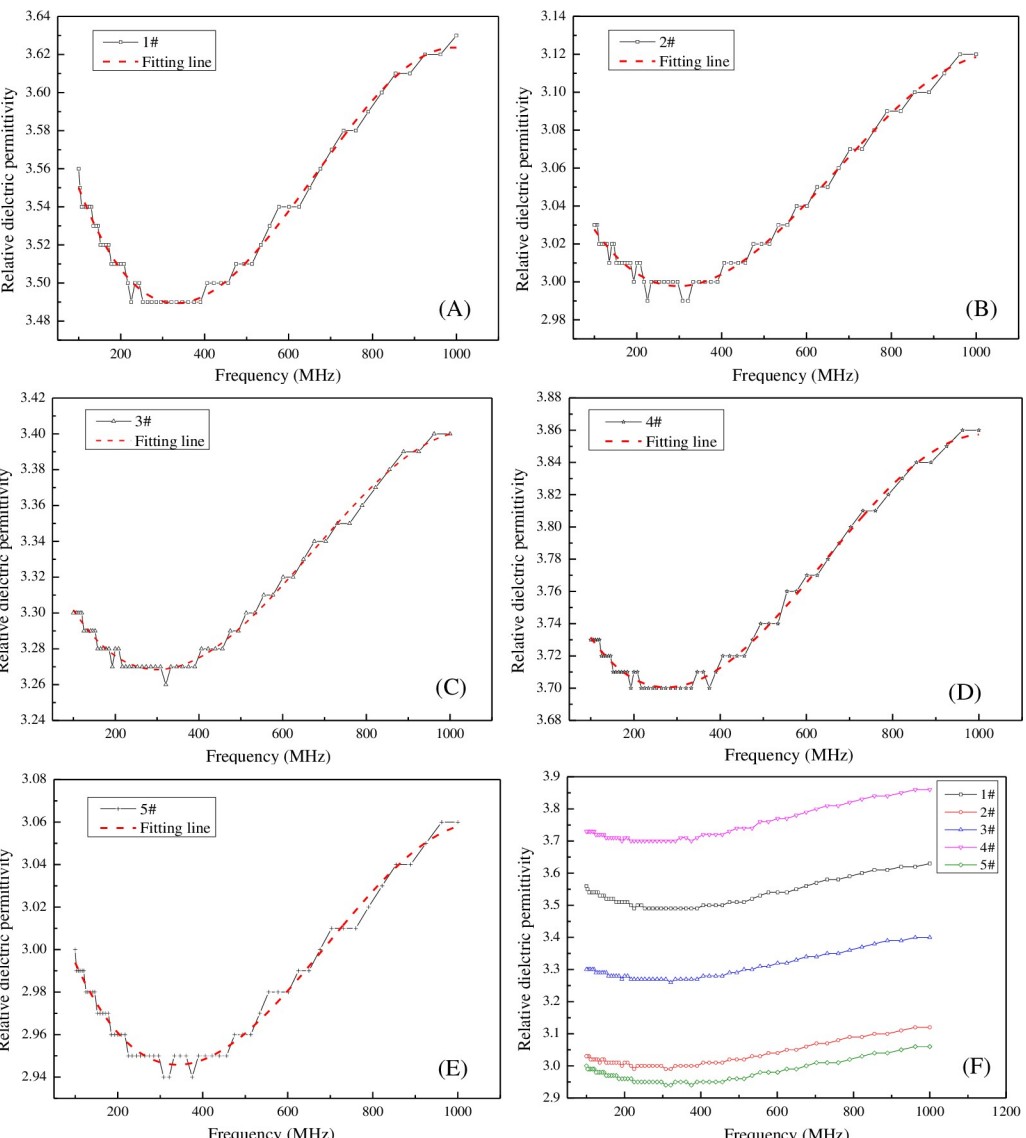

**Fig 4. Changes of the relative dielectric permittivity with frequency for various coals.** A: lignite; B: long flame coal; C: gas coal; D: lean coal; E: anthracite; F: collective plot for all tested samples.

the results are summarized in Table 4.

$$v = \left[ \frac{\varepsilon}{2} \left( \sqrt{1 + (1/2\pi f \varepsilon \rho)^2} + 1 \right) \right]^{-1/2} \qquad (3)$$

where $v$ is the propagation velocity of radar waves in coal (m/s), and $\varepsilon$ is the specific relative dielectric permittivity of coal.

According to the literature [44], the attenuation coefficient of electromagnetic wave propagation in coal could be characterized by Eq (4). Combined with the dielectric constant and electrical conductivity of coal, the variation of attenuation coefficient with frequency could be

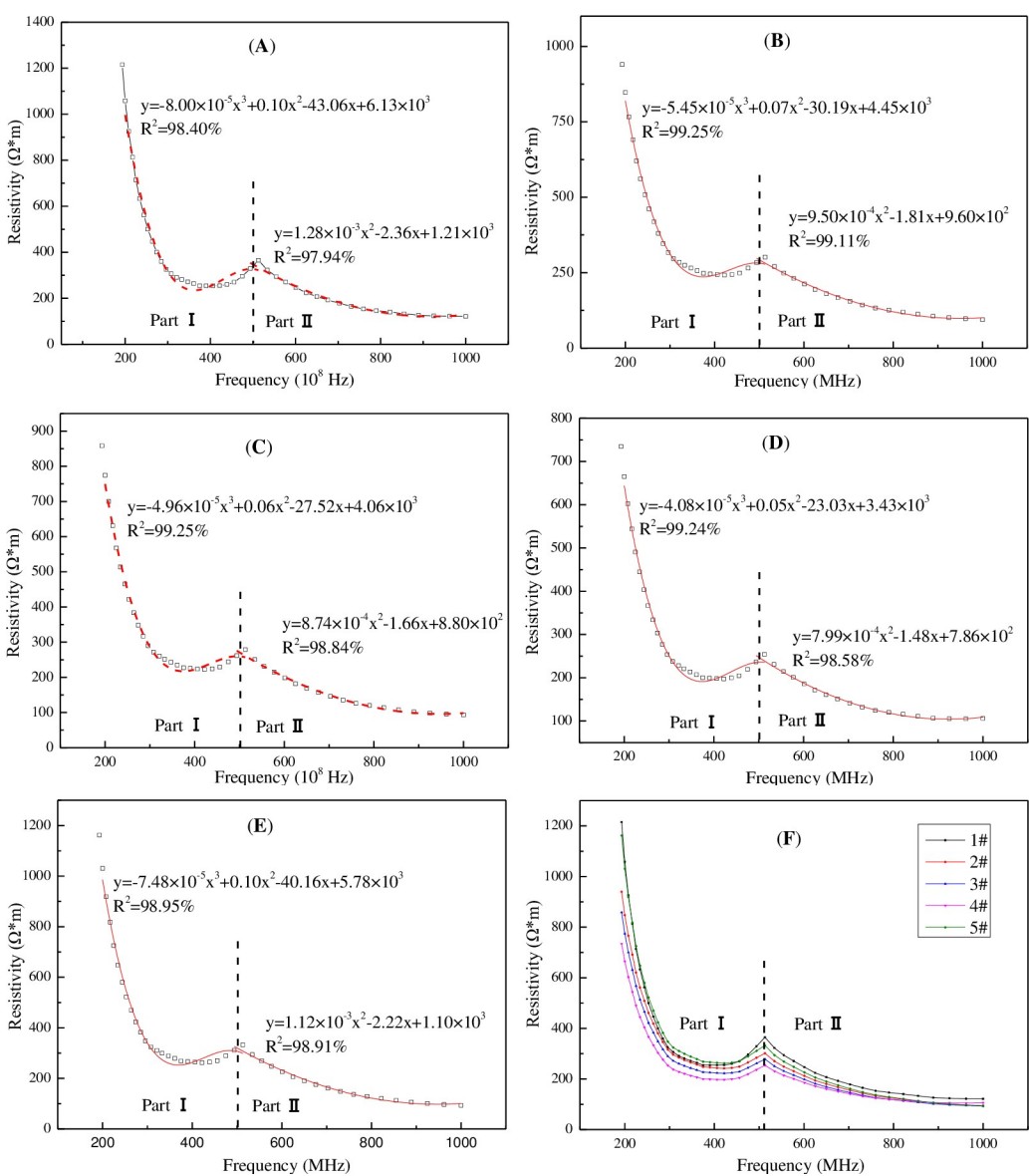

**Fig 5. Variations of tested coals resistivity with frequency.** A: lignite; B: long flame coal; C: gas coal; D: lean coal; E: anthracite; F: collective plot for all tested samples.

**Table 2. Coefficients of fitting equation.**

| Coals | $C_3$ | $C_2$ | $C_1$ | $C_0$ | $R^2$ (%) |
|---|---|---|---|---|---|
| 1# | $-9.24 \times 10^{-10}$ | $1.84 \times 10^{-6}$ | $-9.14 \times 10^{-4}$ | 3.62 | 98.98 |
| 2# | $-5.82 \times 10^{-10}$ | $1.17 \times 10^{-6}$ | $-5.43 \times 10^{-4}$ | 3.07 | 98.77 |
| 3# | $-6.34 \times 10^{-10}$ | $1.28 \times 10^{-6}$ | $-5.96 \times 10^{-4}$ | 3.35 | 99.21 |
| 4# | $-7.90 \times 10^{-10}$ | $1.52 \times 10^{-6}$ | $-6.58 \times 10^{-4}$ | 3.78 | 99.40 |
| 5# | $-6.42 \times 10^{-10}$ | $1.34 \times 10^{-6}$ | $-6.86 \times 10^{-4}$ | 3.05 | 98.49 |

**Table 3. Coefficients of fitting equation.**

| Part | Coefficientntt | 1# | 2# | 3# | 4# | 5# |
|---|---|---|---|---|---|---|
| I | $C_{13}$ | $-8.00\times10^{-5}$ | $-5.45\times10^{-5}$ | $-4.96\times10^{-5}$ | $-4.08\times10^{-5}$ | $-7.48\times10^{-5}$ |
|  | $C_{12}$ | 0.10 | 0.07 | 0.06 | 0.05 | 0.10 |
|  | $C_{11}$ | -43.06 | -30.19 | -27.52 | -23.03 | -40.16 |
|  | $C_{10}$ | $6.13\times10^{3}$ | $4.45\times10^{3}$ | $4.06\times10^{3}$ | $3.43\times10^{3}$ | $5.78\times10^{3}$ |
|  | $R^2$ | 98.40% | 99.25% | 99.25% | 99.24% | 98.95% |
| II | $C_{22}$ | $1.28\times10^{-3}$ | $9.50\times10^{-4}$ | $8.74\times10^{-4}$ | $7.99\times10^{-4}$ | $1.12\times10^{-3}$ |
|  | $C_{21}$ | -2.36 | -1.81 | -1.66 | -1.48 | -2.22 |
|  | $C_{20}$ | $1.21\times10^{3}$ | $9.60\times10^{3}$ | $8.80\times10^{2}$ | $7.86\times10^{2}$ | $1.10\times10^{3}$ |
|  | $R^2$ | 97.94% | 99.11% | 98.84% | 98.58% | 98.91% |

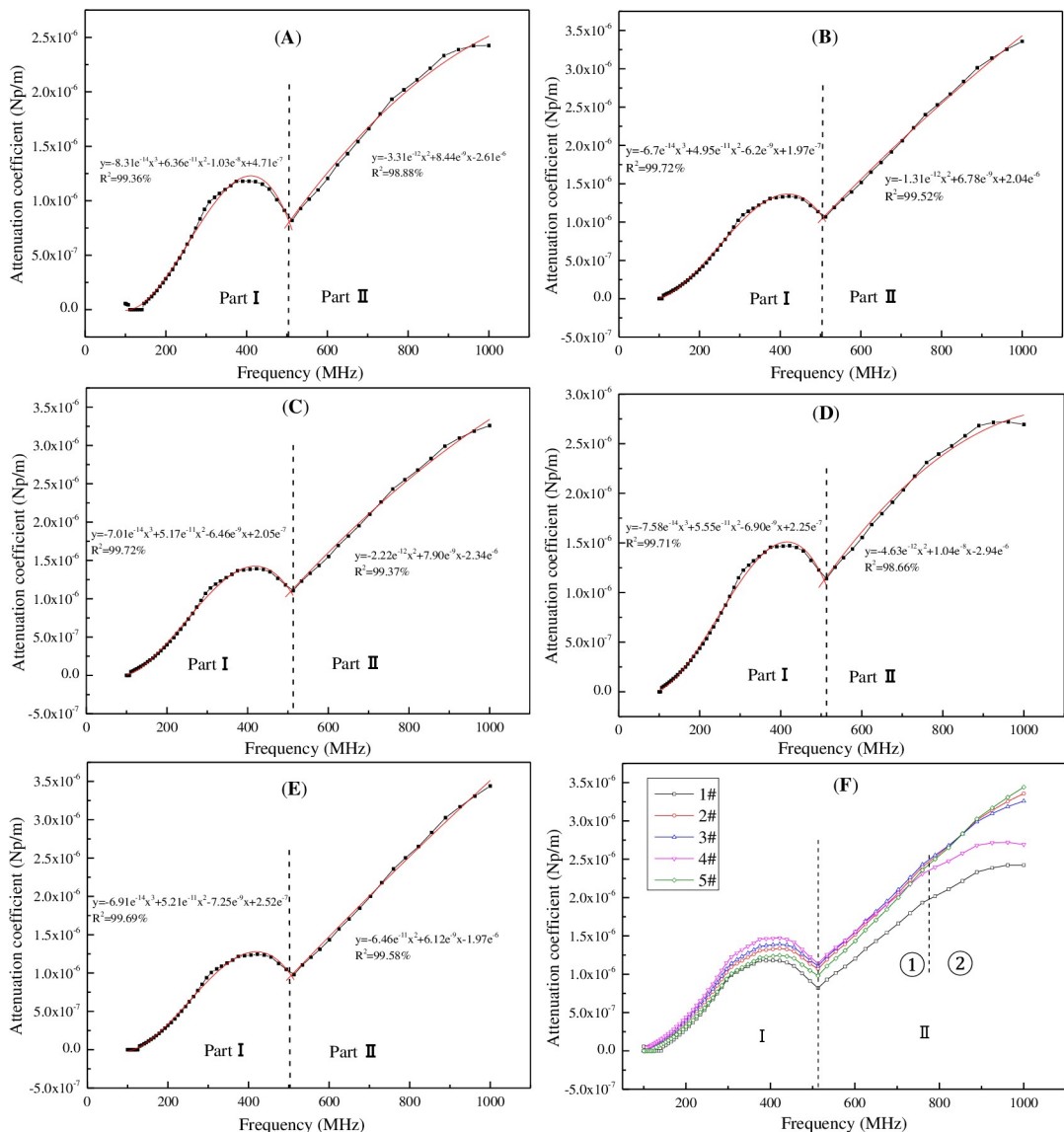

**Fig 6. Changes of attenuation coefficient with frequency.** A: lignite; B: long flame coal; C: gas coal; D: lean coal; E: anthracite; F: collective plot for all tested samples.

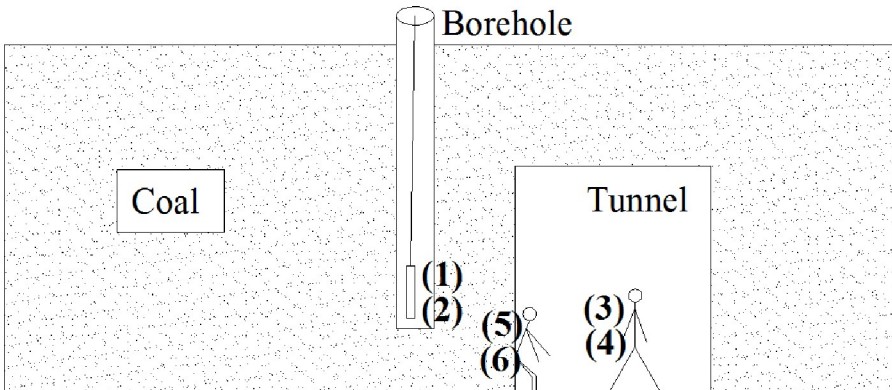

**Fig 7. Schematic diagram of types of reflection and refraction.** Radar waves travel from one object to another. (1): air→coal; (2): coal→air; (3): air→person; (4): person→air; (5): coal→person; (6): person→coal.

obtained (Fig 6).

$$a = 2\pi f \left[ \frac{\varepsilon\mu}{2} \left( \sqrt{1 + (1/2\pi f \rho \varepsilon)^2} - 1 \right) \right]^{1/2} \tag{4}$$

Where $\alpha$ is the attenuation coefficient of radar waves in coal, $f$ is the frequency, $\varepsilon$ is the relative dielectric permittivity of coal, $\rho$ is the specific resistivity of coal, and $\mu$ is the magnetic permeability of coal.

Fig 6 shows the propagation attenuation coefficient of electromagnetic waves in five types of coal followed a similar trend in relation to frequency. It could be divided into two parts, based on the boundary line near 500 MHz (I and II). In part I, the attenuation coefficient was a cubic function of frequency: it increased first and then decreased with the increasing frequency. At the same frequency, attenuation coefficients of coals decrease in the following order: lean coal, gas coal, long flame coal, anthracite, and lignite. In part II, the attenuation coefficient was a quadratic function of frequency: it increased with the increasing frequency. In ① of part II, at the same frequency, the attenuation coefficients decrease in the following order: anthracite, long flame coal (gas coal, lean coal), and brown coal. Among them, the attenuation coefficients of long flame coal, gas coal and lean coal had low differences. In ② of part II, at the same frequency, the attenuation coefficients decreased in the following order: anthracite (long flame coal, gas coal), lean coal, and brown coal. Among them, the attenuation coefficients of anthracite, long flame coal, and gas coal were similar.

The attenuation coefficient and frequency were fitted, and the results are given by in Eq (5) and in Table 5.

$$a = \begin{cases} C'_{13}f^3 + C'_{12}f^2 + C'_{11}f + C'_{10}, \ (f \in [100MHz, 500MHz]) \\ C'_{22}f^2 + C'_{21}f + C'_{20}, \ (f \in (500MHz, 1000MHz]) \end{cases} \tag{5}$$

According to the literature [45], the reflection ($R_{ij}$) and refraction coefficients ($T_{ij}$) of radar waves propagating in coal on the boundary interfaces could be characterized by Eqs (6) and

**Table 4. Propagation speed of radar waves in coal (m/s).**

| Frequency (MHz) | 1# | 2# | 3# | 4# | 5# |
|---|---|---|---|---|---|
| 100 | $15.70\times10^7$ | $17.23\times10^7$ | $16.14\times10^7$ | $15.53\times10^7$ | $17.32\times10^7$ |
| 200 | $16.01\times10^7$ | $17.29\times10^7$ | $16.64\times10^7$ | $15.57\times10^7$ | $17.43\times10^7$ |
| 300 | $16.05\times10^7$ | $17.32\times10^7$ | $16.90\times10^7$ | $15.59\times10^7$ | $17.46\times10^7$ |
| 400 | $16.03\times10^7$ | $17.29\times10^7$ | $16.64\times10^7$ | $15.55\times10^7$ | $17.46\times10^7$ |
| 500 | $16.01\times10^7$ | $17.23\times10^7$ | $16.14\times10^7$ | $15.53\times10^7$ | $17.39\times10^7$ |
| 600 | $15.94\times10^7$ | $17.20\times10^7$ | $16.64\times10^7$ | $15.45\times10^7$ | $17.37\times10^7$ |
| 700 | $15.87\times10^7$ | $17.12\times10^7$ | $16.15\times10^7$ | $15.38\times10^7$ | $17.29\times10^7$ |
| 800 | $15.81\times10^7$ | $17.66\times10^7$ | $16.42\times10^7$ | $15.32\times10^7$ | $17.26\times10^7$ |
| 900 | $15.76\times10^7$ | $17.14\times10^7$ | $16.29\times10^7$ | $15.30\times10^7$ | $17.17\times10^7$ |
| 1000 | $15.74\times10^7$ | $16.84\times10^7$ | $16.26\times10^7$ | $15.26\times10^7$ | $17.14\times10^7$ |

(7), respectively (Fig 7).

$$R_{ij}=\frac{\sqrt{\varepsilon_i}-\sqrt{\varepsilon_j}}{\sqrt{\varepsilon_i}+\sqrt{\varepsilon_j}} \tag{6}$$

$$T_{ij}=\frac{2\sqrt{\varepsilon_i}}{\sqrt{\varepsilon_i}+\sqrt{\varepsilon_j}} \tag{7}$$

where $R_{ij}$ is the reflection coefficient of radar waves at the interface between media $i$ and $j$. $T_{ij}$ is the refraction coefficient of radar waves at the interface between media $i$ and $j$. $\varepsilon_i$ is the relative dielectric constant of medium $i$, and $\varepsilon_j$ is the relative dielectric constant of medium $j$.

The reflection and refraction coefficients of the interfaces were calculated by taking the relative dielectric constant of coal at 500 MHz as an example (Table 6). Clearly, a negative reflection coefficient appears, because the radar wave from the side with a relatively small dielectric constant moves to the side with a relatively high dielectric constant, indicating that the reflected wave was opposite in phase to the incident wave.

The propagation distance of the radar waves in coal was determined by the attenuation coefficient. Eq (4) shows that the independent variables determining the variable $\alpha$ (attenuation coefficient) are $f$ (frequency), $\varepsilon$ (relatively dielectric constant), and $\rho$ (resistivity), i.e., $\alpha=F(f,\varepsilon,\rho)$. From Eq (1), $\varepsilon$ (relatively dielectric constant) is a function of $f$ (frequency), i.e., $f=F^{-1}(\varepsilon)$. From Eq (5), $f$ (frequency) is a function of $\alpha$ (attenuation coefficient), i.e.,

**Table 5. Coefficients of fitting equation.**

| Part | Coefficientntt | 1# | 2# | 3# | 4# | 5# |
|---|---|---|---|---|---|---|
| I | $C'_{13}$ | $-8.31\times10^{-14}$ | $-6.70\times10^{-14}$ | $-7.01\times10^{-14}$ | $-7.58\times10^{-14}$ | $-6.91\times10^{-14}$ |
| | $C'_{12}$ | $6.36\times10^{-11}$ | $4.95\times10^{-11}$ | $5.17\times10^{-11}$ | $5.55\times10^{-11}$ | $5.21\times10^{-11}$ |
| | $C'_{11}$ | $-1.03\times10^{-8}$ | $-6.20\times10^{-9}$ | $-6.46\times10^{-9}$ | $-6.90\times10^{-9}$ | $-7.25\times10^{-9}$ |
| | $C'_{10}$ | $4.71\times10^{-7}$ | $1.97\times10^{-7}$ | $2.05\times10^{-7}$ | $2.25\times10^{-7}$ | $2.52\times10^{-7}$ |
| | $R^2$ | 99.36% | 99.72% | 99.72% | 99.71% | 99.69% |
| II | $C'_{22}$ | $-3.31\times10^{-12}$ | $-1.31\times10^{-12}$ | $-2.22\times10^{-12}$ | $-4.63\times10^{-12}$ | $-6.46\times10^{-11}$ |
| | $C'_{21}$ | $8.44\times10^{-9}$ | $6.78\times10^{-9}$ | $7.90\times10^{-9}$ | $1.04\times10^{-8}$ | $6.12\times10^{-9}$ |
| | $C'_{20}$ | $-2.61\times10^{-6}$ | $2.04\times10^{-6}$ | $-2.34\times10^{-6}$ | $-2.94\times10^{-6}$ | $-1.97\times10^{-6}$ |
| | $R^2$ | 98.88% | 99.52% | 99.37% | 98.66% | 99.58% |

**Table 6. Reflection and refraction coefficients.**

| Coal | Type | (1) air→coal | (2) coal→air | (3) air→person | (4) person→air | (5) coal→person | (6) person→coal |
|------|------|------|------|------|------|------|------|
| 1# | $R_{ij}$ | -0.31 | 0.31 | -0.75 | 0.75 | -0.58 | 0.58 |
|    | $T_{ij}$ | 0.698 | 1.31 | 0.25 | 1.75 | 0.42 | 1.58 |
| 2# | $R_{ij}$ | -0.27 | 0.27 | -0.75 | 0.75 | -0.60 | 0.60 |
|    | $T_{ij}$ | 0.28 | 1.27 | 0.25 | 1.75 | 0.40 | 1.60 |
| 3# | $R_{ij}$ | -0.29 | 0.29 | -0.75 | 0.75 | -0.59 | 0.59 |
|    | $T_{ij}$ | 0.71 | 1.29 | 0.25 | 1.75 | 0.41 | 1.59 |
| 4# | $R_{ij}$ | -0.32 | 0.32 | -0.75 | 0.75 | -0.57 | 0.57 |
|    | $T_{ij}$ | 0.68 | 1.32 | 0.25 | 1.75 | 0.43 | 1.57 |
| 5# | $R_{ij}$ | -0.27 | 0.27 | -0.75 | 0.75 | -0.61 | 0.67 |
|    | $T_{ij}$ | 0.737 | 1.27 | 0.25 | 1.75 | 0.39 | 1.67 |

$\alpha = F^{-1}(f)$. Thus, the propagation attenuation law of radar waves in coal is given by

$$
\begin{cases}
a = 2\pi f \left[ \dfrac{\varepsilon}{2} \left( \sqrt{1 + (1/2\pi f \rho \varepsilon)^2} - 1 \right) \right]^{1/2} \\[2ex]
a = \begin{cases} C'_{13}f^3 + C'_{12}f^2 + C'_{11}f + C'_{10}, & (f \in [100MHz, 500MHz]) \\ C'_{22}f^2 + C'_{21}f + C'_{20}, & (f \in (500MHz, 1.0GHz]) \end{cases} \\[3ex]
\varepsilon = C_3 f^3 + C_2 f^2 + C_1 f + C_0, \quad (f \in [100MHz, 1.0GHz]) \\[3ex]
\rho = \begin{cases} C_{13}f^3 + C_{12}f^2 + C_{11}f + C_{10}, & (f \in [100MHz, 500MHz]) \\ C_{22}f^2 + C_{21}f + C_{20}, & (f \in (500MHz, 1.0GHz]) \end{cases}
\end{cases} \tag{8}
$$

where $\alpha$ is the attenuation coefficient of radar wave propagation in coal; $F$, $\varepsilon$, and $\rho$ represent the frequency, relative dielectric constant, and resistivity, respectively. $C'_{ij}$, $C_i$, $C_{ij}$, ($i, j \in [0, 1, 2, 3]$) are the coefficients of the corresponding function, respectively.

The characteristics of radar waves must be analyzed to find if someone is present behind the coal, when using radar for rescue. The characteristics of the waves were determined by the laws of refraction and reflection, consisting of Eqs (6) and (7):

$$
\begin{cases}
R_{ij} = \dfrac{\sqrt{\varepsilon_i} - \sqrt{\varepsilon_j}}{\sqrt{\varepsilon_i} + \sqrt{\varepsilon_j}} \\[3ex]
T_{ij} = \dfrac{2\sqrt{\varepsilon_i}}{\sqrt{\varepsilon_i} + \sqrt{\varepsilon_j}}
\end{cases} \tag{9}
$$

where $\varepsilon_i$ and $\varepsilon_j$ are the relative dielectric constants of substances $i$ and $j$, respectively. $R_{ij}$ and $T_{ij}$ are the reflection and refraction coefficients, respectively.

Based on the relationship between distance, speed, and time, the distance between the radar and a person in distress can be calculated:

$$\begin{cases} L = v \cdot t \\ v = \left[ \dfrac{\varepsilon}{2} \left( \sqrt{1 + (1/2\pi f \varepsilon \rho)^2} + 1 \right) \right]^{-1/2} \end{cases} \tag{10}$$

where $v$ is the propagation speed of radar waves in coal (m/s).

Eq (11), formed by combining Eqs (8), (9), and (10), is the control equations for the propagation of radar waves in coal:

$$\begin{cases} a = 2\pi f \left[ \dfrac{\varepsilon}{2} \left( \sqrt{1 + (1/2\pi f \rho \varepsilon)^2} - 1 \right) \right]^{1/2} \\[2ex] a = \begin{cases} C'_{13} f^3 + C'_{12} f^2 + C'_{11} f + C'_{10}, & (f \in [100MHz, 500MHz]) \\[1ex] C'_{22} f^2 + C'_{21} f + C'_{20}, & (f \in (500MHz, 1.0GHz]) \end{cases} \\[2ex] \varepsilon = C_3 f^3 + C_2 f^2 + C_1 f + C_0, \ (f \in [100MHz, 1.0GHz]) \\[2ex] \rho = \begin{cases} C_{13} f^3 + C_{12} f^2 + C_{11} f + C_{10}, & (f \in [100MHz, 500MHz]) \\[1ex] C_{22} f^2 + C_{21} f + C_{20}, & (f \in (500MHz, 1.0GHz]) \end{cases} \\[2ex] R_{ij} = \dfrac{\sqrt{\varepsilon_i} - \sqrt{\varepsilon_j}}{\sqrt{\varepsilon_i} + \sqrt{\varepsilon_j}} \\[2ex] T_{ij} = \dfrac{2\sqrt{\varepsilon_i}}{\sqrt{\varepsilon_i} + \sqrt{\varepsilon_j}} \\[2ex] v = \left[ \dfrac{\varepsilon}{2} \left( \sqrt{1 + (1/2\pi f \varepsilon \rho)^2} + 1 \right) \right]^{-1/2} \\[2ex] L = v \cdot t \end{cases} \tag{11}$$

## 3. Results and discussion

Eq (11) could be solved using gprMax code. gprMax is a simulation software based on the finite-difference time-domain theory, which was developed by Dr. Antonis Giannopoulos of the University of Edinburgh [46]. gprMax has been widely used for simulation and modelling of radar wave propagation in GPR applications such as underground pipeline surveys, roadbed inspections, ancient building surveys, military surveys, quality inspections, and hidden grave surveys [47–51]. The simulation calculation process is shown in Fig 8. The simulated physical model is shown in Fig 9(I).

As shown in Fig 9(II), the thickness of a coal body between a trapped person and the radar wave is $L_1$, distance between a person and coal wall is $L_2$, width of the model is $L_3$, length of the model is $L_4$, and dimensions of the human body is 1.6 m ×0.4 m. The specific settings of different models are listed in Table 7. Then, the simulated data was post-processed using MATLAB, and the results are shown in Fig 9(III).

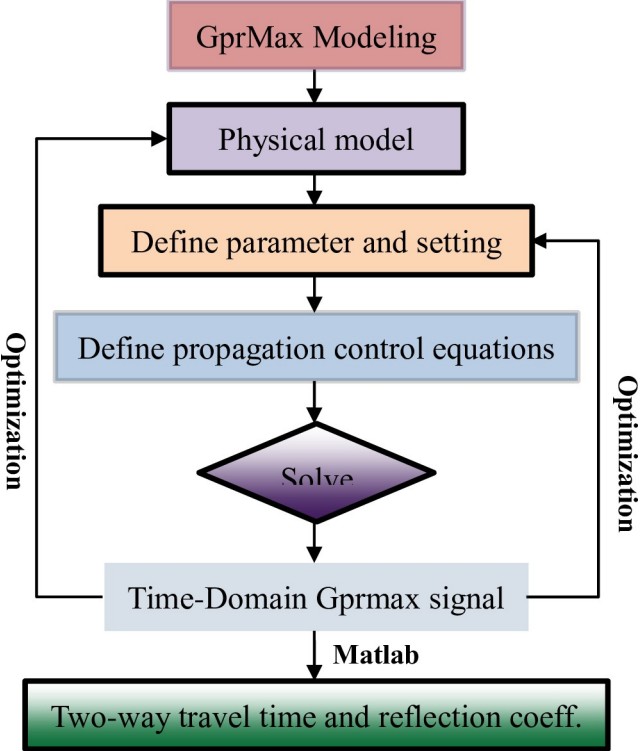

**Fig 8. Simulation calculation flow.**

It was assumed that coal is a semi-infinite continuous space with homogeneous isotropy. The dielectric constant of human body is 50 [52], and that of air is 1; the antenna stepping distance was 0.06 m, and the transmitting and receiving antenna spacing was 0.065 m. The number of measuring lines was 115, spatial grid step was 0.005 m × 0.005 m, permeability of all media in the model was 1.0, boundary condition was a perfect matching layer (PML),

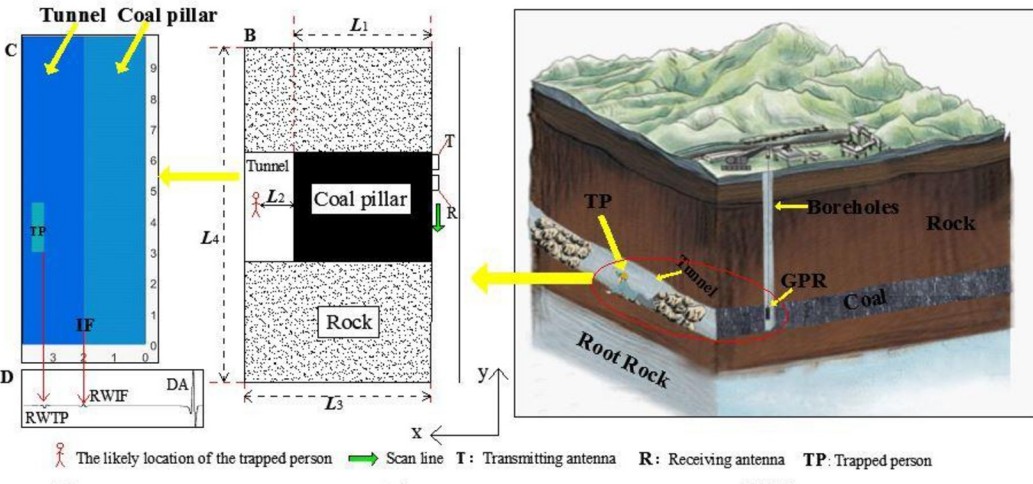

**Fig 9. Geometrical model of rock strata applied in numerical simulation.** I: Schematic illustration of a disaster; II: Schematic physical model; III: Display of physical model in GPR software.

**Table 7. Size of model (m).**

| Model | M1 | M2 | M3 | M4 | M5 | M6 | M7 |
|---|---|---|---|---|---|---|---|
| $L_1$ | 2 | 3 | 4 | 5 | 6 | 7 | 8 |
| $L_2$ | 1.3 | 1.3 | 1.3 | 1.3 | 1.3 | 1.3 | 1.3 |
| $L_3$ | 3.3 | 4.3 | 5.3 | 6.3 | 7.3 | 8.3 | 9.3 |
| $L_4$ | 10 | 10 | 10 | 10 | 10 | 10 | 10 |

excitation function of the radar was Ricker, and the center frequency was 500 MHz. The partial electrical parameters of the model were set as given in Table 8. Due to the large number of graphs of simulation results, only lignite was considered for analysis in this day (Fig 10).

From Fig 10, we found that the amplitudes of the return wave from the interface between the coal pillar and the roadway in seven models were 12.51, 5.48, 2.56, 1.23, 0.63, 0.30, and 0.15 mV/m, and the amplitudes of the return wave reflected from the human body were -20.63, -10.24, -5.30, -2.72, -1.37, -0.69, and -0.35 mV/m. Clearly, the amplitude gradually decreased with the increase in $L_1$ (thickness of coal pillar). The larger the value of $L_1$, the longer the electromagnetic wave travels in coal. Moreover, the longer the electromagnetic wave travels, the higher its energy loss; therefore, the intensity of the target return wave amplitude gradually became smaller. By fitting, it was found that the amplitude of the wave passing through the human body decreases with the increase in $L_1$ as an exponential function (Fig 11). The effective propagation distance of radar waves in coal was 6–8 m for the transducer with the central frequency of 500 MHz.

## 4. Conclusions

The dielectric properties of five different coal samples obtained from various Chinese coal deposits were studied using the alternating current impedance method and the Concept-80 broadband dielectric Spectrum test system for electromagnetic waves, in the frequency range of 100–1000 MHz. Additionally, the propagation mechanism of radar waves in coal was analyzed.

The experimental results showed that the dielectric constants of the five samples follow similar trends. The values of relative dielectric constant of coals decreased first and then increased with the increasing frequency. At the same frequency, the relative permittivity of coal in the descending order was lean coal, lignite, gas coal, long flame coal, and anthracite. The specific resistivity of coal decreased considerably with the increasing frequency, but an upward bump was observed near 500 MHz.

Furthermore, the key parameters of the radar waves propagating through coal were analyzed. The propagation velocities values of radar waves through coals of different ranks and the reflection and refraction coefficients at different interfaces were calculated. The attenuation coefficient of the radar waves in all coal samples was found to follow a similar trend.

With 500 MHz as the dividing point, the relationship between the attenuation coefficient of radar waves and the frequency was divided into two parts. In part I, the frequency was a cubic function of the attenuation coefficient. The attenuation coefficient initially increased to a certain point and then decreased with the increasing frequency. In part II, the frequency was a

**Table 8. Dielectric characterization of coal.**

| Coal | 1# | 2# | 3# | 4# | 5# |
|---|---|---|---|---|---|
| ε | 3.51 | 3.03 | 3.30 | 3.73 | 2.99 |
| ρ (Ω·m) | 328.95 | 290.07 | 267.78 | 242.41 | 318.53 |

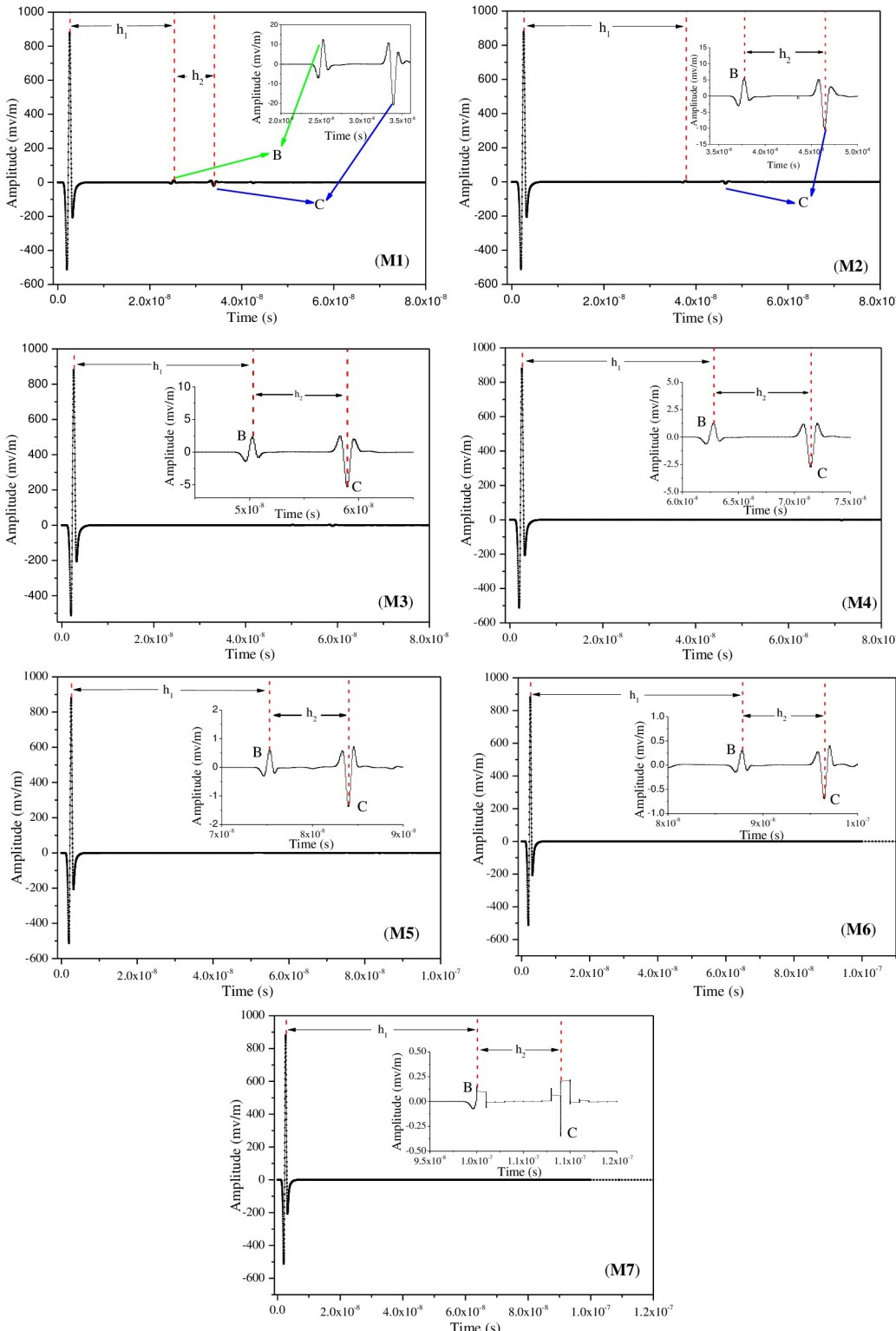

**Fig 10. A plot of signal amplitudes versus travel time.** Mx are the thickness of the coal body between the radar and the trapped person. M1: 2 m; M2: 3 m; M3: 4 m; M4: 5 m; M5: 6 m; M6: 7 m; M7: 8 m. $h_i$ is two-way travel time of radar wave between two objects [52]. The meaning of B and C is given in Fig 9.

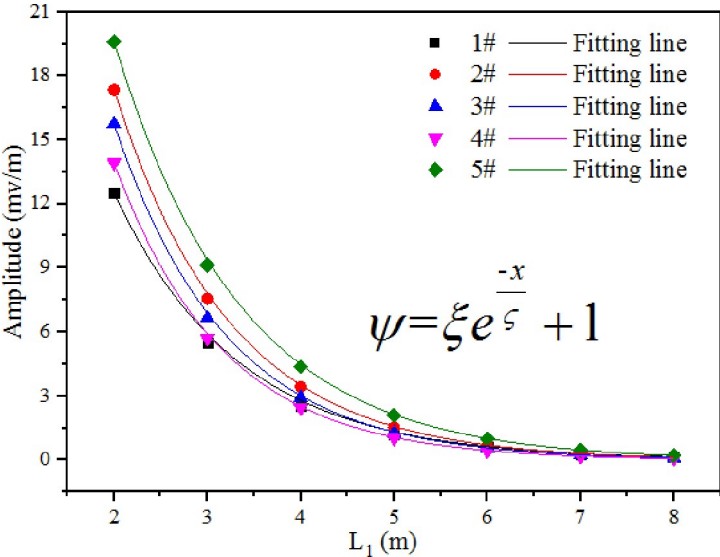

**Fig 11. Relationship between amplitude of a radar signal reflected from a person and a coal seam thickness $L_1$.**

quadratic function of the attenuation coefficient, and the attenuation coefficient increased with the increasing frequency. A set of equations for the propagation of radar waves in coal were established, based on the above findings. By analyzing the governing equations, it was found that the effective penetration distance of radar waves with the dominant frequency of 500 MHz in coal is 6–8 m.

## Supporting information

**S1 File.**
(PDF)

**S2 File.**
(SRD)

## Author Contributions

**Conceptualization:** Duo Zhang.

**Data curation:** Duo Zhang, Rui Tang.

**Formal analysis:** Duo Zhang.

**Funding acquisition:** Duo Zhang.

**Investigation:** Hu Wen.

**Software:** Shixing Fan.

**Writing – original draft:** Duo Zhang.

**Writing – review & editing:** Duo Zhang.

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
