## [Decision Letter · Decision Letter 0]

23 Dec 2019

PONE-D-19-29325

Propagation Law of Ultra-wideband Radar Waves in Coal

PLOS ONE

Dear Dr. Zhang,

Thank you for submitting your manuscript to PLOS ONE. After careful consideration, we feel that it has merit but does not fully meet PLOS ONE’s publication criteria as it currently stands. Therefore, we invite you to submit a revised version of the manuscript that addresses the points raised during the review process.

We would appreciate receiving your revised manuscript by Feb 04 2020 11:59PM. To enhance the reproducibility of your results, we recommend that if applicable you deposit your laboratory protocols in protocols.io, where a protocol can be assigned its own identifier (DOI) such that it can be cited independently in the future. For instructions see: http://journals.plos.org/plosone/s/submission-guidelines#loc-laboratory-protocols

We look forward to receiving your revised manuscript.

Kind regards,

Marco Lepidi, Ph.D.

Additional Editor Comments:

Dear Authors,

two independent reports have been collected for your submission. Both the reports are generally encouraging, but recommend major improvements.

Please, carefully revise your manuscript according to the questions and/or suggestions provided by the reviewers.

Best regards

Marco Lepidi

Journal Requirements:

2. Please include a separate caption for each figure in your manuscript.

Reviewers' comments:

Reviewer's Responses to Questions

**Comments to the Author**

1. Is the manuscript technically sound, and do the data support the conclusions?

Reviewer #1: Partly

Reviewer #2: Partly

2. Has the statistical analysis been performed appropriately and rigorously? 

Reviewer #1: Yes

Reviewer #2: Yes

3. Have the authors made all data underlying the findings in their manuscript fully available?

Reviewer #1: Yes

Reviewer #2: Yes

4. Is the manuscript presented in an intelligible fashion and written in standard English?

Reviewer #1: No

Reviewer #2: No

5. Review Comments to the Author

Reviewer #1: See attached for completed comments.

The key contribution of this paper is that it reports the effects of EM wave frequencies on two fundamental parameters or properties: resistivity and dielectric permitivity for different rank coal samples. I am not quite sure one can claim the outcome/results of the lab testing of these parameter can be declared as “Law”. The claim of the Ultra-wideband radar wave is also arguable as the radar wave used is not necessarily ultra-wideband but the frequency was varied from 100 Mhz – 1000 MHz. Otherwise they would not be able to obtain the parameter variations with frequencies. Based on these measurements, it also derived the EM wave attenuation and speed variations with frequencies for different rank coal samples.

The speed is computed by using the simplified version of the wave speed formula (3). I suggest you may also compute with a more accurate formula such as (Zhou and Fullagar, 2001):

So that it may get different values to the simplified one. Surely the results may do not change much as the frequencies used are relatively high and the approximated version could provide sufficient accuracy.

One of the objectives of the paper is to use GPR for detection of mine trapped personnel after a coal mine disaster. With a frequency ranging from 100 MHz to 1000 MHz, I wonder what kind of penetration capability it will be although the paper claims, in Introduction, the GPR has “strong penetration”. The detection examples in Figures 11 & 12 are simplified too much with no practical use at all. At most, you only illustrate you can estimate the distances of reflection objects without knowing which direction they come from. Such ability is well illustrated in seismic and GPR textbooks. GPR is relatively matured technique. Is there any successful examples such as from controlled experiments in coal mine environment? I guess not. Even if you can receive the reflections from trapped personnel, you need to address the issue such as how to differentiate the human reflections from many other reflections from imhomogeinities of the strata, underground equipment and objects. If you cannot resolve this issue, your GPR measurements have no use to this particular applications.

Based on my above comments, I suggest the author concentrated on the property measurement report rather than try also to address the trapped human detection. Properly understanding the property of coal samples for different frequencies is a good contribution to the literature.

In addition, please improve your English expression of the paper to ensure communicate your finding to your reader effectively. This can be achieved by asking a native English speaker or English Editing services to help you before submit your paper.

Reference

Zhou, B. and Fullagar, P.K., 2001, Delineation of sulphide ore-zones by borehole radar tomography at Hellyer Mine, Australia: Journal of Applied Geophysics, 47, No 3-4, pp.261-269.

Reviewer #2: General remarks: The paper presents data from laboratory testing of electrical properties of various coal samples taken from Chinese mines. Test were performed on a commercial Broadband Dielectric Spectrum Test System -Concept-80. It allows to determine the values of dielectric permittivity and specific resistivity of grinded coal samples subjected to the electromagnetic fields in wide frequency spectrum. On the base of laboratory data a set of approximating equations showing the change of electric properties in frequency domain ranging from 100-1000 MHz was obtained. These equations together with known from EM theory formulas for EM waves speed and attenuation were used for modeling and studying the process of radar waves propagation in a coal seam and reflection from the miners entrapped in collapsed tunnels. The modeling concept assumes that a radar transducer with dominating frequency MHz will be lowered through a vertical or inclined rescue borehole to the depth of coal seam with mining opening. It is assumed that in not collapsed part of tunnel only miner is present. Thus on the path of radar wave from transducer to a human we have only one geological interface reflecting the radar signals (coal-air) and one connected with human body. The human reflector can be located in various distances from the tunnel face. This factor was included in analyzed geometrical model. On the base of modeling results the authors of the paper conclude that a reliable human origin reflected radar signal can be measured when a distance between human and transducer do not exceed 5 m.

Although the concept and research data presented in the manuscript are very interesting and valuable scientifically it needs a thorough revision. There are a lot of deficiencies. Starting from the title, the next parts of manuscript (abstract, introduction and conclusions) in my opinion should be completely re written (my general remarks can be used as a new abstract in the paper). The rest of them need to be corrected. I suggest to remove figures 1 and 2 from the paper. They do not bring nothing and are detached from the content of the paper. Some other figures also need corrections. I recommend to send the manuscript back to authors for major revision. My detailed remarks are included in a word processor file, attached to the revision. I hope that they help the authors to correct the manuscript

6. PLOS authors have the option to publish the peer review history of their article (what does this mean?). If published, this will include your full peer review and any attached files.

Reviewer #1: No

Reviewer #2: No

---

## [Author Response · Author response to Decision Letter 0]

6 Jan 2020

We thank the anonymous reviewers for taking precious time to review our manuscript. We appreciate the reviewers giving us valuable comments that are helpful to improve our manuscript. Here we attempt to reply where and how we revised our manuscript in the light of reviewers’ comments. Revised contents of our manuscript and our replies are marked in red font.

Reviewer #1

Thanks for the reviewer’s comments. In the revised manuscript, we have modified the title as well as removed the word of “Law”. The controversial term UWB to GPR. As shown in Figure 1, we have developed a prototype radar device that can penetrate the coal seam to detect life information. During the research, we tested the dielectric constant of coal and analyzed the propagation of radar waves in the coal. At the initial stage of research, we have assumed some basic physical models. Experts from the China National Key R & D Project team tested our radar prototype at Jining No. 2 Coal Mine. The results show that the radar can penetrate 5m coal seam. We use a combination of the delay band intersection algorithm and the gyroscope at the bottom of the radar to determine the direction of the target. The electrical parameters of strata and human beings are different. Also the directions of reflected waves are different. Therefore, it is possible to distinguish between reflected waves of people and strata. 

Underground equipments and objects bring enormous interference waves during the detection process. We use the following method to screen effective waves. FIR low-pass filter is made use of eliminating direct waves and some background noises. The adaptive minimum mean square algorithm is used to eliminate periodic interference waves and narrowband waves in order to improve the signal-to-noise ratio.

The speed formula was simplified from the accurate formula suggested by the reviewer. We find that the difference between the simplified formula and theaccurate formula was very small. To the certain accuracy, the difference between the results of the two formulas can no longer be seen. For the rigor of the manuscript, we decided to accept the reviewer's opinion that the exact formula was chosen.

The article has been edited by academic editing with the native language.

Reviewer #2

We thank the reviewer carefully checked on this part. The title has been modified to Thank you very much for your detailed suggestions and comments on the manuscript. In particular, the reviewers' general remarks can be directly used as a new abstract of the paper. We have carefully revised the manuscript based on the comments. For example, Figures 1 and 2 have been removed from the paper. We have checked and corrected the words, descriptions, spelling and grammar in the draft in detail.

---

## [Decision Letter · Decision Letter 1]

21 Jan 2020

PONE-D-19-29325R1

Propagation of Ground Penetrating Radar Waves in Chinese Coals

PLOS ONE

Dear Dr. Zhang,

Thank you for submitting your manuscript to PLOS ONE. After careful consideration, we feel that it has merit but does not fully meet PLOS ONE’s publication criteria as it currently stands. Therefore, we invite you to submit a revised version of the manuscript that addresses the points raised during the review process.

We would appreciate receiving your revised manuscript by Mar 06 2020 11:59PM. To enhance the reproducibility of your results, we recommend that if applicable you deposit your laboratory protocols in protocols.io, where a protocol can be assigned its own identifier (DOI) such that it can be cited independently in the future. For instructions see: http://journals.plos.org/plosone/s/submission-guidelines#loc-laboratory-protocols

We look forward to receiving your revised manuscript.

Kind regards,

Marco Lepidi

Marco Lepidi, PhD

Associate Professor, Academic Editor of PLOS ONE

Dipartimento di Ingegneria Civile, Chimica ed Ambientale

Università degli Studi di Genova (Italy)

Mail: Marco.Lepidi@unige.it

Website: http://www3.dicca.unige.it/mlepidi/index.html

Google Scholar: http://scholar.google.it/citations?user=uePAdVUAAAAJ&hl=it

Additional Editor Comments (if provided):

Dear Authors,

two reports have been collected for your submission from the same reviewers of the first round. Both the reports are generally positive, but one of the report still recommends major improvements.

Please, take this last opportunity to carefully revise your manuscript according to the questions and/or suggestions provided by the reviewers.

Best regards

Reviewers' comments:

Reviewer's Responses to Questions

**Comments to the Author**

1. If the authors have adequately addressed your comments raised in a previous round of review and you feel that this manuscript is now acceptable for publication, you may indicate that here to bypass the “Comments to the Author” section, enter your conflict of interest statement in the “Confidential to Editor” section, and submit your "Accept" recommendation.

Reviewer #1: (No Response)

Reviewer #2: All comments have been addressed

2. Is the manuscript technically sound, and do the data support the conclusions?

Reviewer #1: Partly

Reviewer #2: Yes

3. Has the statistical analysis been performed appropriately and rigorously? 

Reviewer #1: Yes

Reviewer #2: Yes

4. Have the authors made all data underlying the findings in their manuscript fully available?

Reviewer #1: Yes

Reviewer #2: Yes

5. Is the manuscript presented in an intelligible fashion and written in standard English?

Reviewer #1: No

Reviewer #2: No

6. Review Comments to the Author

Reviewer #1: My concern on how to differentiate the radar reflections from many other reflections from inhomogeneities of the strata, underground equipment and objects has not been properly addressed. I fully understand that human is different from strata and equipment. But there are so many situations that you can observe similar reflections to those from human bodies. For example, will you be able to tell the difference of human reflections from the waterbody reflections in the strata? This is related to multi-solutions of geophysical problems – the same observation can be caused by many different situations. In addition, I have not been convinced by the authors’ response that their prototype radar system with a gyro can tell the direction of received radar waves, unless their radar system is a directional system. Can you provide an example of the testing results for trapped humans using your prototype radar system. Otherwise, there is a 360 degrees ambiguity. If you want to use the radar method for detection of the trapped miners in disasters, you have to address and discuss these issues.

I don’t think your paper has been properly edited by an academic editing people as there are many Chinglish expressions remained – some of them are in the attached marked manuscript. It is interesting to see that you just copied and pasted the reviewer’s comments as your abstract without changing a word.

Reviewer #2: The manuscript has been deeply improved and all my suggestions have been considered. Nevertheless, I've noticed some minor errors or "bugs" which have to be removed before publication. I attach a revised manuscript indicating noticed "bugs". After removal of them the manuscript can be accepted for publishing.

7. PLOS authors have the option to publish the peer review history of their article (what does this mean?). If published, this will include your full peer review and any attached files.

Reviewer #1: No

Reviewer #2: No

---

## [Author Response · Author response to Decision Letter 1]

23 Feb 2020

The author has responded to all reviewer questions in "Response to reviewers".

---

## [Decision Letter · Decision Letter 2]

16 Mar 2020

PONE-D-19-29325R2

Propagation of Ground Penetrating Radar Waves in Chinese Coals

PLOS ONE

Dear Dr. Zhang,

Thank you for submitting your manuscript to PLOS ONE. After careful consideration, we feel that it has merit but does not fully meet PLOS ONE’s publication criteria as it currently stands. Therefore, we invite you to submit a revised version of the manuscript that addresses the points raised during the review process.

We would appreciate receiving your revised manuscript by Apr 30 2020 11:59PM. To enhance the reproducibility of your results, we recommend that if applicable you deposit your laboratory protocols in protocols.io, where a protocol can be assigned its own identifier (DOI) such that it can be cited independently in the future. For instructions see: http://journals.plos.org/plosone/s/submission-guidelines#loc-laboratory-protocols

We look forward to receiving your revised manuscript.

Kind regards,

Marco Lepidi

Marco Lepidi, PhD

Associate Professor, Academic Editor of PLOS ONE

Dipartimento di Ingegneria Civile, Chimica ed Ambientale

Università degli Studi di Genova (Italy)

Mail: Marco.Lepidi@unige.it

Website: http://www3.dicca.unige.it/mlepidi/index.html

Google Scholar: http://scholar.google.it/citations?user=uePAdVUAAAAJ&hl=it

XXXXXXXXXXXXXXXXXXXXXXXXXXXXXXXXXXXXXXXXXXXXXXXXXXXXXXXXXXXXXXXXXXXXXXXXXX

Dear Authors,

The two reviewers have reconsidered your submission. One of them is fully satisfied, the other is still asking for some minor improvements.

Please, reply (and/or carefully revise your manuscript according) to the questions and/or suggestions provided by the second reviewers.

Best regards

Marco Lepidi

Reviewers' comments:

Reviewer's Responses to Questions

**Comments to the Author**

1. If the authors have adequately addressed your comments raised in a previous round of review and you feel that this manuscript is now acceptable for publication, you may indicate that here to bypass the “Comments to the Author” section, enter your conflict of interest statement in the “Confidential to Editor” section, and submit your "Accept" recommendation.

Reviewer #1: (No Response)

Reviewer #2: All comments have been addressed

2. Is the manuscript technically sound, and do the data support the conclusions?

Reviewer #1: Partly

Reviewer #2: Yes

3. Has the statistical analysis been performed appropriately and rigorously? 

Reviewer #1: No

Reviewer #2: Yes

4. Have the authors made all data underlying the findings in their manuscript fully available?

Reviewer #1: Yes

Reviewer #2: Yes

5. Is the manuscript presented in an intelligible fashion and written in standard English?

Reviewer #1: No

Reviewer #2: No

6. Review Comments to the Author

Reviewer #1: See attached comments

I don’t think the authors have addressed my previous comments properly.

The authors replied

1> We can distinguish the return wave is reflected by the human or the formation water. There are two reasons for this. The first is that the electrical parameters of water are significantly different from the electrical parameters of the human body. The second is that the human's reflected waves are loaded with regular heartbeat signals.

Theoretically what you said is correct. However, in practice, how do you differentiate the human reflection from the formation water body reflection from recorded radargrams? Please provide your method in the paper. Can the radar record show or detect human’s heartbeats? What is the resolution of the GPR system? Can you please provide any evidence or references on how your GPR can detect heartbeating? Please write your paper more scientifically, not with statements without support.

2> As the reviewer said, our radar system is a directional system. As shown in Figure 1, the radar structure includes the transmit and receive window. The size of the window is 70 mm × 50 mm. Experts from the China National Key R & D Project team tested our radar prototype at Jining No. 2 Coal Mine. One of the test scenarios is shown in Figure 2, and the test results are shown in Figure 3.

You only provide numerical examples which prove nothing as any different objects can cause radar reflections theoretically. Your Figure 3 in your reply also shows nothing – why are there no actual radar data presented to support your conclusion? How good is the directionality of your radar system? What is the radiation pattern? Why don’t you include a real data example in this paper for human detection or event to see the heartbeating?

3> The article has been edited by academic editing with the native language. We have provided the certificate of english editing in the previous reply. If required, we can invite academic editors to make it more standard. After reading the reviewers' comments, we all think that it is very perfect. It is more suitable as an abstract for a dissertation. Therefore, we have completely used the reviewer's comments as a new abstract.

I have made some suggestions in the last review but you totally ignored my suggestions and believe that your English is fine! You may pay some attention to the following comments/suggestions (I am sure there are more in the paper) :

1. Line 29: Prefer to change “the authors of the paper conclude” to “, it is concluded”. The reviewer’s writing is ok, but I don’t think the authors of the paper should use similar expression.

2. Line 31: Not quite sure what exact you mean by “do not exceed 5-8 m”! Do you mean it should not exceed 5m or you really mean it should not exceed 8m. Or you mean the distances between 5m and 8m are ok. How did you derive these number? Or is this because you only modelled the data up to 8m?

3. Line 43: I am not sure you have used “particularly significant” and “major” correctly for these events – please check your original Chinese meaning with the corresponding English.

4. Line 67/8: “Fan and Jia et al.” -> “Fan et al.”

5. Line 69: “Fan and Chang et al.” -> “Fan et al.”

6. Line 82: “Peng” -> “Peng et al.”

7. Line 85: “Xu” -> “Xu et al.”

8. Line 88: “Emslie et al.” -> “Emslie and Lagace”

9. Line 94: “Luo and Stolarczyk et al.” -> “Luo et al.”

10. Line 106: “Dielectric properties measurement” ->”Dielectric property measurement”

11. Line 108-110: Change “in” to “from”.

12. Line 130: “followed similar regularities” -> “follow a similar trend”. Using “regularity” is very Chinese!

13. Line 139: “larger” -> “higher”. The constant is “higher” not “larger”.

14. Line 160: “large” -> “high”. We normally say the attenuation is high or low not large or small in English!

15. Line 184, 188, 191: “large”->”high”; “small” ->”low”.

Reviewer #2: I would response in point 5 "yes" but I have noticed some, following typographical errors in the manuscript:

line 102 ........ radar waves in coal "are" constructed (should be "is" constructed),

line 132 ........ largest "one" (should be plural "ones"),

line 216 ......... is relatively (should be "relative").

Figure captions:

Fig.5 variations of tesed coals (should be "tested"),

Fig.9 I - schematic "of" illustration of disaster (should be schematic illustration of disaster),

Fig.9 II - Schematic "of simplified" of physical model (should be schematic physical model),

Fig.11 - Relationship between (or a plot) between............. Dual expression. Should be only one eg. Relationship between amplitudes........ (remove "or a plot").

Notice to editor:

Although the current title of manuscript has been my proposal (revision 1) I suggest the editor to consider the following change "A concept of borehole radar for rescuing miners trapped in deep coal mines". After revisions made this title reflects better the content of the paper. In my opinion it could also increase the interest in the article. I leave decision upon the editor.

7. PLOS authors have the option to publish the peer review history of their article (what does this mean?). If published, this will include your full peer review and any attached files.

Reviewer #1: No

Reviewer #2: No

---

## [Author Response · Author response to Decision Letter 2]

22 Apr 2020

Reviewer #1

1.Thanks for the reviewer’s comments. The beating of the heart can cause micro-undulation vibrations in the chest cavity at approximately periodic frequencies. The purpose of human body detection can be achieved form identifying the frequency of these micro-motion signals. Obstacles such as formation water bodies have no periodic frequency characteristics. But it still causes interference to radar waves. Therefore, it is necessary to effectively process the radar echoes in order to extract and identify signals containing human micro-motion information features. The method of identifying the characteristics of the micro-motion signal of radar echo information is as follows. The first is the preprocessing of the echo signal. It mainly includes DC offset removal, background removal, time-varying gain, and mean denoising of the echo signal. The second is the selection of sliding time window function and energy integration. The third is to use wavelet transform to process the energy integration results. The number of radar signal sampling points is 4096, and the scanning speed is 16Hz. In particular, radar can be used to obtain heartbeat and respiration signals, which has been employed in many areas. Such as: “Xiaolin Liang, Hao Zhang, Shengbo Ye, et al. Improved denoising method for through-wall vital sign detection using UWB impulse radar, Digital Signal Processing 74(2018) 72–93”, “Minhhuy Le. Heart rate extraction based on eigenvalues using UWB impulse radar remote sensing, Sensors and Actuators A 303 (2020) 111689”, “F. JalaliBidgoli, S. Moghadami, S. Ardalan, A compact portable microwave life-detection device for finding survivors, IEEE Embed. Syst. Lett. 8(1) (2016) 10–13”.

2. We thank the reviewer for this valuable comment. He is correct that any different object can cause radar reflections in theory. However, the slight fluctuation of the chest cavity caused by heartbeats are periodic and can be identified from radar echoes through a series of algorithm processing. In this article, we focus on establishing an approximate equation of radar wave propagation in coal and try to solve it. So there are no examples of measured data in this article. As a response to this comment, we add a set of actual radar data in the attachment. The schematic diagram of the experiment is shown in Figure 1 (In the file of "Response to reviewers"). In future works, we will research on measured data and try to propose micro-vibration recognition algorithms for the life information. The detection direction of the radar system is shown in Figure 2 (In the file of "Response to reviewers"). The angle between the detection direction and the horizontal direction is smaller than 120 oC, while the angle between the detection direction and the vertical direction is smaller than 90 oC. According to the detection angle, this detection scheme can be realised by adjusting direction of the radar window.

3.We thank the reviewer’s comments. We found a lot of problems about English expression. We improved it according to the suggestion. The article was edited again by academic editing with the native language. Thank you again for your meaningful suggestions and comments.

Reviewer #2

We thank the reviewer carefully checked on this part. Thank you very much for your detailed suggestions and comments on the manuscript. We have carefully revised the manuscript based on the comments. Thank you again for your meaningful suggestions and comments.

---

## [Editor Report · Decision Letter 3]

6 May 2020

Propagation of Ground Penetrating Radar Waves in Chinese Coals

PONE-D-19-29325R3

Dear Dr. Zhang,

We are pleased to inform you that your manuscript has been judged scientifically suitable for publication and will be formally accepted for publication once it complies with all outstanding technical requirements.

With kind regards,

Marco Lepidi, Ph.D.

Marco Lepidi, PhD

Associate Professor, Academic Editor of PLOS ONE

Dipartimento di Ingegneria Civile, Chimica ed Ambientale

Università degli Studi di Genova (Italy)

Mail: Marco.Lepidi@unige.it

Website: http://www3.dicca.unige.it/mlepidi/index.html

---

## [Editor Report · Acceptance letter]

8 May 2020

PONE-D-19-29325R3 

Propagation of Ground Penetrating Radar Waves in Chinese Coals 

Dear Dr. Zhang:

I am pleased to inform you that your manuscript has been deemed suitable for publication in PLOS ONE. Congratulations! Your manuscript is now with our production department. 

With kind regards,

on behalf of

Professor Marco Lepidi 

Academic Editor

PLOS ONE